# Effect of Infraorbital and/or Infratrochlear Nerve Blocks on Postoperative Care in Patients with Septorhinoplasty: A Meta-Analysis

**DOI:** 10.3390/medicina59091659

**Published:** 2023-09-14

**Authors:** Do Hyun Kim, Jun-Beom Park, Sung Won Kim, Gulnaz Stybayeva, Se Hwan Hwang

**Affiliations:** 1Department of Otolaryngology-Head and Neck Surgery, Seoul St. Mary’s Hospital, College of Medicine, The Catholic University of Korea, Seoul 06591, Republic of Korea; dohyuni9292@naver.com (D.H.K.); kswcfb@ut.ac.kr (S.W.K.); 2Department of Periodontics, College of Medicine, The Catholic University of Korea, Seoul 06591, Republic of Korea; jbassoon@catholic.ac.kr; 3Department of Physiology and Biomedical Engineering, Mayo Clinic, Rochester, MN 55905, USA; stybayeva.gulnaz@mayo.edu; 4Department of Otolaryngology-Head and Neck Surgery, Bucheon St. Mary’s Hospital, College of Medicine, The Catholic University of Korea, Seoul 06591, Republic of Korea

**Keywords:** nerve block, nasal septum, rhinoplasty, pain, meta-analysis

## Abstract

*Background and Objectives*: Through a comprehensive meta-analysis of the pertinent literature, this study evaluated the utility and efficacy of perioperative infraorbital and/or infratrochlear nerve blocks in reducing postoperative pain and related morbidities in patients undergoing septorhinoplasty. *Materials and Methods*: We reviewed studies retrieved from the PubMed, SCOPUS, Embase, Web of Science, and Cochrane databases up to August 2023. The analysis included a selection of seven articles that compared a treatment group receiving perioperative infraorbital and/or infratrochlear nerve blocks with a control group that either received a placebo or no treatment. The evaluated outcomes covered parameters such as postoperative pain, the amount and frequency of analgesic medication administration, the incidence of postoperative nausea and vomiting, as well as the manifestation of emergence agitation. *Results*: The treatment group displayed a significant reduction in postoperative pain (mean difference = −1.7236 [−2.6825; −0.7646], I^2^ = 98.8%), as well as a significant decrease in both the amount (standardized mean difference = −2.4629 [−3.8042; −1.1216], I^2^ = 93.0%) and frequency (odds ratio = 0.3584 [0.1383; 0.9287], I^2^ = 59.7%) of analgesic medication use compared to the control. The incidence of emergence agitation (odds ratio = 0.2040 [0.0907; 0.4590], I^2^ = 0.0%) was notably lower in the treatment group. The incidence of postoperative nausea and vomiting (odds ratio = 0.5393 [0.1309; 2.2218], I^2^ = 60.4%) showed a trend towards reduction, although it was not statistically significant. While no adverse effects reaching statistical significance were reported in the analyzed studies, hematoma (proportional rate = 0.2133 [0.0905; 0.4250], I^2^ = 76.9%) and edema (proportional rate = 0.1935 [0.1048; 0.3296], I^2^ = 57.2%) after blocks appeared at rates of approximately 20%. *Conclusions*: Infraorbital and/or infratrochlear nerve blocks for septorhinoplasty effectively reduce postoperative pain and emergence agitation without notable adverse outcomes.

## 1. Introduction

Septorhinoplasty is a surgical procedure designed to improve the aesthetic features of the nasal region and correct any deformities. The postoperative phase can often involve significant discomfort, which arises from factors such as soft tissue trauma, irritation of periosteal tissues, and the effects of interventions on bones, including osteotomies [1,2]. Moreover, pain can be exacerbated by nasal packing after the surgery. Common postoperative complications include pain, edema, and periorbital ecchymosis [2,3]. There is a consensus that employing localized pain-management techniques postoperatively can lead to fewer complications and decreased overall costs [4]. The American Society of Anesthesiologists’ guidelines for acute pain management during the perioperative stage recommend a multimodal analgesia approach, emphasizing the use of regional blockade techniques, where applicable [4]. While opioids are effective for managing postoperative pain, their use can lead to undesirable side effects such as sedation, respiratory depression, and episodes of nausea and vomiting [2,5]. Moreover, such potential adverse effects can hinder the patient’s timely discharge from care [5]. This has led to an increasing emphasis on the application of peripheral nerve blocks during postoperative pain management. Some propositions even advocate for regional analgesia techniques to be the primary means of pain relief after plastic surgery [6]. Peripheral nerve blocks offer multiple advantages, such as reducing tissue edema, ensuring a more comprehensive range of anesthesia, and diminishing pain at the surgical site [7].

The infraorbital nerve, a branch of the maxillary division of the trigeminal nerve, plays a pivotal role in providing sensory innervation to the cutaneous areas around the nose and the nasal septum [7]. The infratrochlear nerve innervates the skin of the dorsum of the upper part and both sides of the nose. These two peripheral nerve blocks have been reported to facilitate pain management, reduce complications, and reduce anesthetic agent consumption after nasal procedures. Conducting a nerve block of this nerve has been shown to promote effective pain control, reduce complications, and decrease the requirement for anesthetic agents following nasal surgeries [8,9,10,11,12,13,14].

However, the effect of infraorbital and/or infratrochlear nerve blocks on the peri-nasal region remains a topic of debate [8,15]. We hypothesized that infraorbital and/or infratrochlear nerve blocks would be effective because septorhinoplasty is accompanied by significant postoperative pain. Therefore, we conducted a systematic review and meta-analysis to evaluate the influence of infraorbital and/or infratrochlear nerve blocks on postoperative care in patients undergoing septorhinoplasty.

## 2. Materials and Methods

This systematic review and meta-analysis was conducted in accordance with the guidelines set by the Preferred Reporting Items for Systematic Reviews and Meta-Analyses (PRISMA) [16]. The research protocol was proactively registered in the Open Science Framework (Charlottesville, VA, USA), accessible at the URL: https://osf.io/r3jve/ (accessed on 13 August 2023).

### 2.1. Search Strategy and Study Selection

The criteria, based on Population, Intervention, Comparison, Outcomes, and Study (PICOS), are as follows: (1) Population: patients who underwent septorhinoplasty; (2) Intervention: perioperative infraorbital and/or infratrochlear nerve blocks; (3) Comparison: either a placebo or no treatment; (4) Outcomes: grading of postoperative pain as reported by patients, quantity, and frequency of administered analgesic drugs, instances of postoperative nausea and vomiting and emergence agitation, and any events of adverse effects related to the infraorbital and/or infratrochlear nerve blocks; (5) Study design: not specified. Clinical studies up until August 2023 were sourced from PubMed, SCOPUS, Google Scholar, Embase, and the Cochrane Register of Controlled Trials. Key search terms employed included ‘nerve block’, ‘rhinoplasty’, ‘septorhinoplasty’, ‘infraorbital nerve’, ‘infratrochlear nerve’, ‘pain’, ‘emergency agitation’, ‘nausea’, and ‘adverse effect.’ Detailed search terms and queries are listed in Appendix A. A librarian with more than a decade of experience facilitated the database searches, and the authors further examined references in the identified articles to ensure no omissions. Two independent reviewers (DHK and JP) meticulously assessed the titles and abstracts of potential studies, eliminating those considered irrelevant to perioperative infraorbital and/or infratrochlear nerve blocks. If abstracts did not provide enough information for a clear decision, the full texts of those studies were reviewed. Any disagreements between the two were settled by consulting a third reviewer (SHH).

To qualify for the review, studies had to meet specific criteria: they needed to involve patients undergoing septorhinoplasty and the application of an intraoperative infraorbital block. Studies focusing on additional procedures, such as sinus surgery, were excluded, ensuring the emphasis remained on septoplasty. Any research not offering clear, measurable data related to the outcomes of interest, or where extracting valuable information from the published results was impractical, was not considered. Figure 1 provides a visual representation of the search strategy used to pinpoint the studies included in this meta-analysis.

### 2.2. Data Extraction and Risk of Bias Assessment

Data from qualifying studies were extracted using standardized data collection forms [17,18,19]. The parameters assessed included the intensity of patient-reported postoperative pain, the amount and regularity of analgesic drug administration, the onset of postoperative nausea and vomiting, emergence agitation, and any reported adverse effects related to the infraorbital and/or infratrochlear nerve blocks. A comparison was made between the treatment group and the control group (those who received either no treatment or a saline injection) during the perioperative phase.

The data chosen for analysis encompassed patient demographics, pain intensity ratings given by patients, specifics regarding analgesic drug usage in terms of amount and regularity, rates of postoperative nausea and vomiting, events of emergence agitation, occurrences of side effects, and the stated *p*-values denoting the contrast between the treatment and control groups, as outlined in the chosen studies. This thorough data aggregation aimed to uncover the potential influence of the nerve block on postoperative complications and adverse effects.

### 2.3. Statistical Analyses

A statistical evaluation of the included studies was performed using the R-4.3.1 program developed by the R Software 4.3.1 Foundation based in Vienna, Austria. For quantitative variables, the meta-analysis was performed using either the standardized mean difference (SMD) or mean difference (MD). Employing SMD as a summary statistic allowed for the alignment of study outcomes onto a consistent scale, particularly when evaluations of similar outcomes were achieved using varied methods. This approach was chosen to gauge the quantity of administered analgesic medication, due to the lack of a universal scale across all studies. MD, indicating the average variance between the treatment and control groups, was calculated when the outcomes across studies were consistent and conveyed in similar units within the patient-reported pain grading scale. For metrics such as the regularity of analgesic medication administration, the incidence of postoperative nausea and vomiting, events of emergence agitation, and recorded side effects, the odds ratio (OR) was determined.

Heterogeneity was evaluated using the I^2^ test, which describes the degree of variance across studies attributable to reasons beyond random chance, with values ranging from 0 (indicating no heterogeneity) to 100 (representing maximum heterogeneity). All reported findings were paired with a 95% confidence interval (CI), and *p*-values were cited for two-tailed tests. If significant heterogeneity in outcomes was detected, indicated by I^2^ > 50, the random-effects model, following the DerSimonian–Laird method, was adopted. This model assumes different variances in true treatment effects among studies and anticipates a normal distribution of these. Where needed, subgroup analysis was also undertaken. Situations where heterogeneity was not significant, indicated by I^2^ < 50, were examined using the fixed-effects model. This model, leveraging the inverse variance method, assumes a common source for all studies within the evaluated population.

To identify possible publication bias, both a funnel plot and Egger’s test were utilized. The Duval and Tweedie’s trim-and-fill method was also applied to adjust the combined effect size, accounting for any perceived publication bias. Moreover, sensitivity analyses were conducted to gauge the impact of individual studies on the overall meta-analysis results, which involved performing repeated meta-analyses, each time excluding one specific study.

## 3. Results

We evaluated a total of seven studies encompassing 414 participants. Detailed information about the individual studies is provided in Table 1, while the results of bias assessments can be found in Table 2 and Table 3. Utilizing both Egger’s test and Begg’s funnel plot, specifically for assessing pain scores, we determined that there was no discernible publication bias among the incorporated studies (*p* = 0.4912). However, it is worth noting that an assessment of publication bias for other outcomes was not conducted due to the limited number of studies available, which made the creation of a funnel plot infeasible.

### 3.1. Effect of Preoperative Infraorbital and/or Infratrochlear Nerve Blocks on Patient-Reported Pain Score and the Quantity and Frequency of Administered Analgesic Medication Compared to the Control Group

The postoperative pain score (MD = −1.7236 [−2.6825; −0.7646], I^2^ = 98.8%), and quantity (SMD = −2.4629 [−3.8042; −1.1216], I^2^ = 93.0%) and frequency (OR = 0.3584 [0.1383; 0.9287], I^2^ = 59.7%) of administered analgesics, in the treatment group were markedly lower than the values in the control group, as shown in Figure 2. Notably, significant inter-study heterogeneity (I^2^ > 50%) was observed in the mentioned outcomes.

The overall analysis did not differentiate based on who decided the quantity of analgesics administered (whether patient self-controlled or clinician-controlled), the interval after surgery when the pain score was assessed (within 2 h, 2–8 h, 8–24 h, or after 2 days), or the criteria for administering analgesics (a pain score > 3 or >7). This lack of differentiation likely contributed to the high heterogeneity (>50%) observed across all studies’ results.

In a subgroup analysis that evaluated postoperative pain relative to the elapsed time, no significant variances were observed among the different time intervals (within 2 h: −1.5693 [−3.3455; 0.2068], I^2^ = 98.9%; 2–8 h: 2.4546 [−5.0657; 0.1566], I^2^ = 99.3%; 8–24 h: −1.6588 [−3.3991; 0.0815], I^2^ = 98.6%; 2 days: −0.5000 [−0.8686; −0.1314], I^2^ = NA) (*p* = 0.1895). These findings suggest that infraorbital and/or infratrochlear nerve blocks might maintain their efficacy up to 2 days postoperatively.

In a subgroup analysis that assessed the amount of analgesics administered based on the decision maker (patient vs. clinician), the quantity of analgesics self-administered by patients (−4.3254 [−5.1226; −3.5281], I^2^ = NA) was significantly less than the amount decided by clinicians (−1.8263 [−2.6487; −1.0040], I^2^ = 76.5%) (*p* < 0.0001). It is worth noting that postoperative pain intensity typically peaks within the first 24 h following surgery, with this pain often intensifying during the evening [13]. While only a single study utilized patient-controlled analgesics, the findings might indicate that nerve blocks can improve the patient’s postoperative pain experience.

In a separate subgroup analysis that focused on the frequency of analgesic administration based on the criteria for administering them (either a pain score > 3 or >7), there were no significant differences between the two thresholds (pain score > 3: 0.4604 [0.1666; 1.2720], I^2^ = 53.4%; >7: 0.2032 [0.0251; 1.6485], I^2^ = 71.8%) (*p* = 0.4910). This suggests that nerve blocks might be equally effective for mitigating both moderate and severe postoperative pain.

### 3.2. Effect of Preoperative Infraorbital and/or Infratrochlear Nerve Blocks on the Incidence of Postoperative Nausea and Vomiting, Emergence Agitation, and Occurrence of Side Effects Compared to the Control Group

The use of infraorbital and/or infratrochlear nerve blocks led to a significant reduction in the occurrence of emergence agitation (OR = 0.2040 [0.0907; 0.4590], I^2^ = 0.0%) compared to the control group (Figure 3). Although the decrease in postoperative nausea and vomiting due to the infraorbital and/or infratrochlear nerve blocks did not achieve statistical significance, it is important to note a trend towards its reduction compared to the control group (OR = 0.5393 [0.1309; 2.2218], I^2^ = 60.4%). Significant heterogeneity was evident among the analyzed studies (I^2^ > 50%) regarding the incidence of postoperative nausea and vomiting. This analysis did not differentiate the specific post-surgery time frames when assessing postoperative nausea and vomiting, such as during the immediate Post-Anesthesia Care Unit phase or within the 24 h that followed. This exclusion likely contributed to the notable heterogeneity (>50%) seen in the overall findings of the reviewed studies.

In a subgroup analysis that focused on the time of assessment (whether in the Post-Anesthesia Care Unit or within 24 h), no significant differences were identified (Post-Anesthesia Care Unit: 0.5397 [0.0173; 16.8663], I^2^ = 58.9%, 24 h: 0.5515 [0.0942; 3.2278], I^2^ = 73.9%) (*p* = 0.9913). This suggests that the beneficial effects of the infraorbital and/or infratrochlear nerve blocks in preventing postoperative complications persist up to 24 h after the operation.

Regarding side effects, no major adverse reactions, including neurological deficits, were highlighted in the studies included. However, incidences of hematoma (proportional rate = 0.2133 [0.0905; 0.4250], I^2^ = 76.9%) and edema (proportional rate = 0.1935 [0.1048; 0.3296], I^2^ = 57.2%) following nerve blocks were observed at a rate of ~20%.

### 3.3. Sensitivity Analysis

An iterative sensitivity assessment was performed, wherein individual studies were consecutively excluded from the meta-analysis. No single study was found to significantly impact the overall trend.

## 4. Discussion

Our meta-analysis demonstrates that perioperative pain management utilizing infraorbital and/or infratrochlear nerve blocks effectively reduced postoperative pain and the usage of analgesic drugs. Moreover, there was a significant decline in the incidence of emergence agitation. Importantly, there were no notable adverse effects, including neurological deficits, reported.

Postoperative pain is characterized as acute inflammatory pain that originates with surgical trauma and typically resolves in tandem with tissue healing. When pain triggers a release of catecholamines, it can precipitate cardiovascular incidents, undesirable neuroendocrine or metabolic changes, thromboembolic events, pulmonary complications, and prolonged hospital stays [20]. It is crucial to efficiently address postoperative pain as it aids rapid mobilization, ensures adequate intake of fluids and nutrition, and accelerates the return to normal physical activities [21]. After surgical procedures, pain becomes a significant factor influencing patient well-being. Achieving comfort in the immediate postoperative stage plays a substantial role in enhancing patient contentment and overall satisfaction. Alongside conventional early analgesic approaches, the introduction of additional interventions that aim to reduce the need for analgesics and improve patient well-being is of utmost importance [22,23].

Emergence agitation is a postanesthetic phenomenon marked by psychomotor behaviors such as agitation, confusion, disorientation, and potential aggressive conduct that occurs during recovery from general anesthesia [24,25]. This state can pose significant challenges for patients, leading to complications such as injuries, bleeding, amplified pain sensations, accidental self-extubation, and the unintentional removal of catheters. Beyond the direct risks to patients, emergence agitation can also have negative ramifications for healthcare staff and may result in increased hospital costs [26]. Its occurrence has particularly been highlighted in relation to septorhinoplasty [8]. To counter this, adept pain management techniques have been proposed to reduce its incidence [8].

Nonetheless, achieving effective analgesic management remains a significant challenge. Postoperative pain affects a vast segment of the global population. The postoperative period inherently carries an increased likelihood of morbidity and mortality, with some cases potentially attributed to the use of analgesic agents [27,28,29]. Securing adequate analgesia through the use of regional nerve blocks, opioid analgesics, and nonsteroidal anti-inflammatory drugs has proven effective for both preventing and mitigating agitation, even for procedures traditionally considered to lack significant pain stimuli [30]. However, resorting to pharmacological preemptive measures can inadvertently lead to prolonged sedation and ensuing hemodynamic changes, which may include hypotension, hypertension, and bradycardia. These events can correlate with extended durations in the Post-Anesthesia Care Unit and a subsequent increase in overall hospital costs [8]. Furthermore, anesthetic variables play a crucial role in the onset of postoperative nausea and vomiting. These variables include the use of inhalational anesthetics, the length of anesthesia, the subsequent use of postoperative opioid analgesics, and the introduction of nitrous oxide [31]. A relationship between the administration of opioids during the postoperative period and an increased likelihood of postoperative nausea and vomiting has been suggested. This relationship appears to be dose-dependent [32].

Consequently, many patients often undergo treatment using a mix of non-opioid analgesic agents, a strategy known as multimodal analgesia [21]. The main goal of this approach is to achieve a synergistic or additive beneficial effect while reducing individual analgesic doses. This not only helps in preventing adverse effects but also reduces dependence on opioids and the associated range of opioid-related side effects [33,34]. Notable among the non-opioid drugs included in multimodal analgesic strategies are paracetamol, nonsteroidal anti-inflammatory drugs (NSAIDs), corticosteroids, ketamine, local anesthetics, and gabapentinoids [35]. Currently, various combinations of non-opioid analgesic drugs are employed in clinical settings [34]. In a forward-looking clinical study, a comparative evaluation assessed postoperative pain in patients undergoing specific otorhinolaryngologic surgeries with local anesthesia combined with sedation. Interestingly, the research revealed a significant difference, with patients who had septorhinoplasty experiencing significantly more pain compared to those who had septoplasty [36]. This observation aligns with another separate study where patients who had septoplasty showed little to no need for postoperative pain relief, maintaining effective pain management throughout the assessed postoperative phase [37]. Moreover, in the said study, patients who had undergone comprehensive nasal correction surgery displayed a steady improvement in pain scores, matching the levels seen in post-septoplasty patients just 6 days after the procedure. Taken together, these findings underscore the importance of a focused strategy to mitigate perioperative pain related to septorhinoplasty.

Peripheral nerve blocks involve the injection of a local anesthetic near the nerve that serves the surgical area. These anesthetics work by changing the sodium permeability of cell membranes, effectively halting nerve impulse transmission and leading to pain relief [38]. A distinctive feature of peripheral nerve blocks is their tendency to produce fewer side effects and complications, such as reduced swelling at the surgical site and lessened pain perception. When considering the sensory nerves relevant to septorhinoplasty surgery, the infraorbital and/or infratrochlear nerve blocks stand out in the facial region [39,40].

Our findings suggest that infraorbital and/or infratrochlear nerve blocks administered to patients undergoing septorhinoplasty provide effective pain control with minimal complications and reduce the reliance on perioperative opioids. Furthermore, the use of ultrasound guidance may increase the efficiency and safety of the procedure [41,42]. It is essential to highlight the significance of addressing agitation due to its potential detrimental effects on both the patient and the surgical site [43]. In addition, early post-surgical trauma can negatively influence the surgical results, which is particularly concerning in procedures such as rhinoplasty that involve modifications of intricate nasal bones. Consequently, adopting a cautious approach to minimize the risks posed by trauma induced by agitation is crucial, particularly for patients subjected to these surgeries [11]. While the precise efficiency of infraorbital and/or infratrochlear nerve blocks in countering agitation is still under investigation, previous studies, such as that by Choi et al. [8], indicate a strong link between pain and emergence agitation. Our observations echo this trend, with systemic magnesium administration being shown to reduce agitation during the recovery phase, which resonates with the known pharmacokinetics of magnesium. Moreover, the group that received nerve blocks experienced effective pain management throughout the postoperative phase.

Historically, efforts to alleviate emergence agitation have mainly centered on pharmacological solutions. Drugs such as ketamine, dexmedetomidine, and propofol have been found to be effective for managing emergence agitation in adults. However, it is worth noting that relying on pharmacological prevention can lead to prolonged sedation and hemodynamic changes, including hypotension, hypertension, and bradycardia. Such outcomes could result in extended durations in the Post-Anesthesia Care Unit, prompting considerations about cost-efficiency.

In the context of our research, we primarily concentrated on pain control using peripheral nerve blocks. As such, we champion the adoption of a multimodal pain management strategy, integrating systemic analgesics with peripheral nerve blocks, as the recommended method for septorhinoplasty when appropriate. Additionally, when performing septorhinoplasty, office-based local anesthesia can be facilitated by pain control caused by nerve blocks, and clinical data on this will also be needed.

However, this study is not without its limitations. Despite thorough subgroup analyses, fully addressing the heterogeneity regarding the effect of the perioperative nerve blocks on procedural morbidities was challenging. Several factors contribute to these challenges. First, the use of various local anesthetic agents in different studies, including levobupivacaine, ropivacaine, and bupivacaine, introduces variability. It is essential to note that the clinical characteristics of levobupivacaine and ropivacaine are similar to those of racemic bupivacaine, with differences mainly related to slight variation in anesthetic potency. Specifically, racemic bupivacaine displays higher potency compared to levobupivacaine and ropivacaine [44]. Second, we also included studies in which an infraorbital nerve block was performed but not an infratrochlear nerve block. This may contribute to the increased heterogeneity. It is critical to understand that the infraorbital nerve provides sensory innervation to the nasal skin and septum mobile nasi, whereas the infratrochlear nerve is responsible for the innervation of the nasal root [8]. In the future, additional clinical research will be needed on the effectiveness of individual nerve blocks for pain control after septorhinoplasty. The combined effect of varied anesthetic agents and the potential addition of an infratrochlear nerve block might be a primary source of the observed heterogeneity. Finally, another source of variability is the innate differences in peripheral nerve block techniques as executed by different clinicians. Each clinician’s unique application approach can introduce variability, further adding to the heterogeneity noted in our findings.

## 5. Conclusions

This study underscores the efficacy of infraorbital and/or infratrochlear nerve blocks for septorhinoplasty in mitigating postoperative pain and emergence agitation. Furthermore, the application of this intervention did not present significant adverse outcomes, such as neurological deficits.

## Figures and Tables

**Figure 1 medicina-59-01659-f001:**
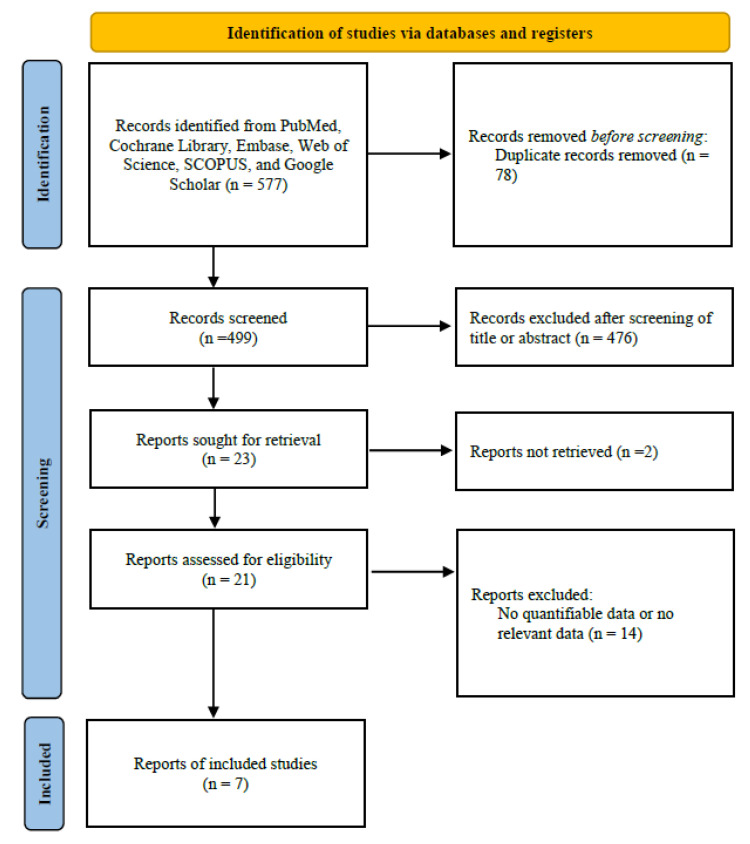
Diagram of study selection.

**Figure 2 medicina-59-01659-f002:**
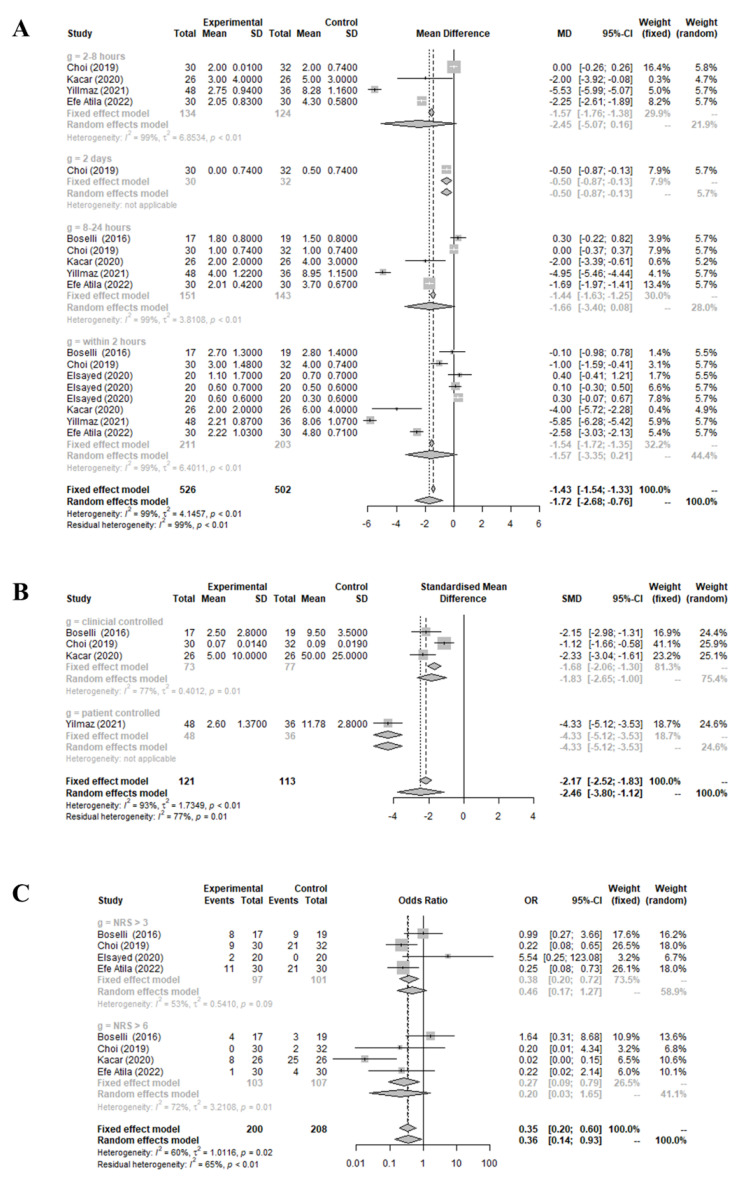
Comparison of preoperative infraorbital and/or infratrochlear nerve blocks versus placebo: mean difference in postoperative pain score (**A**), standard mean difference in amount of analgesic used (**B**), and odds ratio for frequency of analgesic drug use (**C**) [8,9,10,11,12,13].

**Figure 3 medicina-59-01659-f003:**
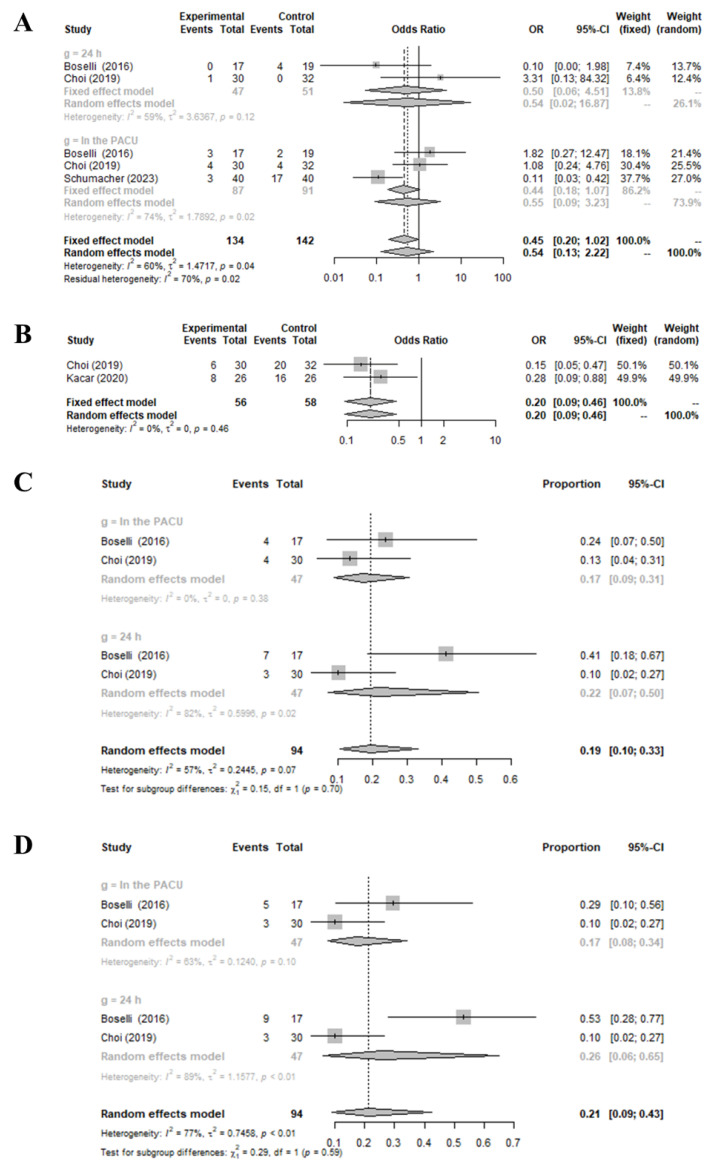
Comparison of preoperative infraorbital and/or infratrochlear nerve blocks versus placebo: odds ratio for incidence of postoperative nausea and vomiting (**A**) and emergence agitation (**B**), with incidence rates for edema (**C**) and hematoma (**D**) [8,9,11,14].

**Table 1 medicina-59-01659-t001:** Summary of studies included in the meta-analysis.

Study (Year)	Study Design	Number of Patients	Sex (Male/Female)	Age, Median (Range) or Mean (SD), y	Nation	Procedure	Medication	Anesthesia	Medication	Outcomes
Choi (2019) [8]	RCT	62	NA	21.97 ± 1.47	Korea	bilateral infraorbital and infratrochlear nerve blocks	0.5% ropivacaine	G/A	Isotonic saline	Postoperative pain score, used analgesic amount, frequency of used analgesic drug, incidence of postoperative nausea and vomiting, incidence of emergence agitation, incidence rate of edema and hematoma
Boselli (2016) [9]	RCT	36	13/23	38 ± 14	France	bilateral infraorbital and infratrochlear nerve blocks	10 mL of 0.25% levobupivacaine	G/A	Isotonic saline	Postoperative pain score, used analgesic amount, frequency of used analgesic drug, incidence of postoperative nausea and vomiting, incidence rate of edema and hematoma
Kacar (2020) [11]	RCT	52	24/28	27.38 ± 7.09	Turkey	bilateral infraorbital and infratrochlear nerve blocks	4 mL of 0.5% bupivacaine	G/A	G/A only	Postoperative pain score, used analgesic amount, frequency of used analgesic drug, incidence of emergence agitation
Elsayed (2020) [10]	Observational Study	40	26/14	26 (8)	Saudi Arabia	bilateral infraorbital and infratrochlear nerve blocks	5 mL of 0.25% levobupivacaine with 5 mL of diluted adrenaline 1:10,000	G/A	G/A only	Frequency of used analgesic drug
Yılmaz (2021) [12]	Observational Study	84	26/58	26.88 ± 5.45	Turkey	bilateral infraorbital and supraorbital nerve blocks	5 mg bupivacaine and 10 mg lidocaine for a total of 1.5 mL	G/A	G/A only	Postoperative pain score, used analgesic amount
Efe Atila (2022) [13]	Observational Study	60	27/33	30.5 ± 4	Turkey	bilateral infraorbital nerve blocks	15 mg bupivacaine hydrochloride to infraorbital foramen	G/A	G/A only	Postoperative pain score, frequency of used analgesic drug
Schumacher (2023) [14]	Observational Study	80	NA	27.6	USA	bilateral infraorbital nerve blocks	1.5 mL preoperative and postoperative bupivacaine	NA	NA	Incidence of postoperative nausea and vomiting

RCT, randomized controlled trials; G/A, general anesthesia; NA, not available.

**Table 2 medicina-59-01659-t002:** Quality of individual non-randomized controlled trial methodology.

Study	Selection ^a^				Comparability ^b^		Exposure ^c^			Newcastle–Ottawa Scale Score
	1	2	3	4	5A	5B	6	7	8	
Elsayed (2020) [10]	Yes	No	Yes	Yes	No	No	Yes	Yes	Yes	6
Yılmaz (2021) [12]	Yes	No	No	Yes	Yes	Yes	Yes	Yes	Yes	7
Efe Atila (2022) [13]	Yes	No	Yes	Yes	No	No	Yes	Yes	Yes	6
Schumacher (2023) [14]	Yes	Yes	Yes	Yes	No	No	Yes	Yes	Yes	7

A star rating system was used to indicate the quality of a study, with a maximum rating of nine stars. A study could be awarded a maximum of one star for each numbered item within the selection and exposure categories. ^a^: Selection (4 items): adequacy of case definition; representativeness of cases; selection of controls; and definition of controls. ^b^: Comparability (1 item): comparability of cases and controls on the basis of design or analysis. ^c^: Exposure (3 items): ascertainment of exposure; same method of ascertainment used for cases and controls; and non-response rate (same rate for both groups).

**Table 3 medicina-59-01659-t003:** Individual randomized controlled trial methodological quality.

Study	Random Sequence Generation	Allocation Concealment	Blinding of Participants and Personnel	Blinding of Outcome Assessment	Incomplete Outcome Data Addressed	Free of Selective Reporting	Risk of Bias of Randomized Studies
Choi (2019) [8]	Yes	Yes	Yes	Yes	Yes	Yes	Low
Boselli (2016) [9]	Yes	Yes	Yes	No	Yes	Yes	Low
Kacar (2020) [11]	Yes	Yes	Yes	No	Yes	Yes	High

## Data Availability

The raw data of individual articles used in this meta-analysis are included in the main text or Appendix A.

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
