# Peer review of "Effect of Infraorbital and/or Infratrochlear Nerve Blocks on Postoperative Care in Patients with Septorhinoplasty: A Meta-Analysis"

_medicina, 2023, doi:10.3390/medicina59091659_

Round 1

Reviewer 1 Report

Dear Authors,

You present a very complex meta analysis regarding the use of nerve blocks for controlling pain associated to nasal septum procedures.

However there are some aspects that require your attention.

You either add in the title the reference to infratrochlear or supraorbital nerve blocks, or you exclude the studies which are not focusing on the infraorbital block alone.

Also clarify if the patients were operated under general anesthesia or with local anesthesia.

Is this study the basis for using such nerve blocks on a wider scale in order to transform the septorhinoplasty into an in office procedure?

Were the nerve blocks performed under ultrasound guidance? In the discussion section mention also that the anesthetic could be infiltrated using sonography of the paranasal sinuses. Reference this to the work by Neagos, A., Dumitru, M., Vrinceanu, D., Costache, A., Marinescu, A. N., & Cergan, R. (2021). Ultrasonography used in the diagnosis of chronic rhinosinusitis: From experimental imaging to clinical practice. Experimental and therapeutic medicine21(6), 611. https://doi.org/10.3892/etm.2021.10043

You should detail the in the conclusion the need for further development of a clinical study focusing only on infraorbital nerve block for pain management in septorhinoplasty.

Looking forward to receiving your improved manuscript.

Author Response

You either add in the title the reference to infratrochlear or supraorbital nerve blocks, or you exclude the studies which are not focusing on the infraorbital block alone.

â—Ž Reply:

According to the reviewer’s comment, we have revised the title and related content to "infraorbital and/or infratrochlear nerve blocks".

Also clarify if the patients were operated under general anesthesia or with local anesthesia.

â—Ž Reply:

Of the 8 studies, 7 were general anesthesia, and 1 study was not mentioned. We have added this to Table 1.

Is this study the basis for using such nerve blocks on a wider scale in order to transform the septorhinoplasty into an in office procedure?

â—Ž Reply:

Nerve blocks have been mentioned to be used to reduce pain after septorhinoplasty. It might be used to convert to office-based local anesthesia, but there are no studies conducted with local anesthesia. However, as the reviewer commented, the pain control due to nerve blocks can facilitate office-based local anesthesia. We added this point to the discussion section (Page 14, Line 351 to 353).

Were the nerve blocks performed under ultrasound guidance? In the discussion section mention also that the anesthetic could be infiltrated using sonography of the paranasal sinuses. Reference this to the work by Neagos, A., Dumitru, M., Vrinceanu, D., Costache, A., Marinescu, A. N., & Cergan, R. (2021). Ultrasonography used in the diagnosis of chronic rhinosinusitis: From experimental imaging to clinical practice. Experimental and therapeutic medicine, 21(6), 611. https://doi.org/10.3892/etm.2021.10043

â—Ž Reply:

All included studies were not conducted with ultrasound guidance. However, as the reviewer pointed out, the use of ultrasound guidance may increase the efficacy and safety of the procedure (Michalek, P.;Donaldson,W.;McAleavey, F.; Johnston, P.; Kiska, R.Ultrasound imaging of the infraorbital foramen and simulation of the ultrasound-guided infraorbital nerve block using a skull model. Surg. Radiol. Anat. 2013, 35, 319–322). We added this content to the main text along with the reference mentioned by the reviewer (Page 13, Line 326 to 327; reference 41 and 42).

You should detail the in the conclusion the need for further development of a clinical study focusing only on infraorbital nerve block for pain management in septorhinoplasty.

â—Ž Reply:

We revised the sentences as follows (Page 14, Line 362 to 368).

We also included studies in which an infraorbital nerve block was performed but not a infratrochlear nerve block. This may contribute to increased heterogeneity. It is critical to understand that the infraorbital nerve provides sensory innervation to the nasal skin and septum mobile nasi, whereas the infratrochlear nerve is responsible for the innervation of the nasal root [8]. In the future, additional clinical research will be needed on the effectiveness of individual nerve blocks for pain control after septorhinoplasty.

Reviewer 2 Report

Dear colleagues!

The meta-analysis is well done, but I would like to emphasize a few points.

1. Need to add a null hypothesis

2. If possible, include a graphic abstract

3. As we know, it is difficult to objectively assess pain. Why did you not take into account the works devoted specifically to the objectification of the effectiveness of anesthesia (registration of somatosensory evoked potentials of the brain, pain thresholds, etc.). Finally, why is there no section on subjective assessment - there is more of this information, for example, a visual analogue scale.

4. The list of references is old and contains many references, which are more than 15 years old. Now there are new modern and more effective anesthetics, keep this in mind when working with text.

Author Response

  1. Need to add a null hypothesis

â—Ž Reply:

We added the following hypothesis to the introduction section (Page 3, Line 68 to 71):

We hypothesized that infraorbital nerve block could be effective because septorhinoplasty is accompanied by significant postoperative pain.

  1. If possible, include a graphic abstract

â—Ž Reply:

Because this manuscript is a systematic review and meta-analysis, there are limitations in applying a graphic abstract.

  1. As we know, it is difficult to objectively assess pain. Why did you not take into account the works devoted specifically to the objectification of the effectiveness of anesthesia (registration of somatosensory evoked potentials of the brain, pain thresholds, etc.). Finally, why is there no section on subjective assessment - there is more of this information, for example, a visual analogue scale.

â—Ž Reply:

As the reviewer mentioned, postoperative pain is subjective and difficult to evaluate simply by pain alone. Therefore, we added not only the postoperative pain score but also the frequency of used analgesic drugs, incidence of postoperative nausea and vomiting, and incidence of emergence agitation additionally presented in the reports as outcome data. Because the objectification of the effectiveness of anesthesia was not performed in the included studies, a meta-analysis could not be performed.

  1. The list of references is old and contains many references, which are more than 15 years old. Now there are new modern and more effective anesthetics, keep this in mind when working with text.

â—Ž Reply:

According to the reviewer’s comment, we have replaced the content with the latest reference (reference 4).